# High-Salt Diet and Intestinal Microbiota: Influence on Cardiovascular Disease and Inflammatory Bowel Disease

**DOI:** 10.3390/biology13090674

**Published:** 2024-08-29

**Authors:** Xueyang Wang, Fuyuan Lang, Dan Liu

**Affiliations:** 1Queen Mary College, Nanchang University, Xuefu Road, Nanchang 330001, China; xueyang.wang@se21.qmul.ac.uk (X.W.); f.lang@se21.qmul.ac.uk (F.L.); 2School of Pharmacy, Jiangxi Medical College, Nanchang University, Nanchang 330006, China

**Keywords:** high-salt diet, intestinal microbiota, inflammatory bowel disease, cardiovascular disease

## Abstract

**Simple Summary:**

Salt, or sodium chloride, is a crucial component of a balanced diet. However, an excessive consumption of salt can lead to a range of adverse health effects that are often overlooked. Salt enters the body’s digestive system through the ingestion of everyday foods and exerts an effect on a key component of the digestive system, namely the intestinal microbiota. Some studies have indicated that alterations in the composition of the intestinal microbiota may influence the functioning of the human body. This paper presents a review of the effects of a high-salt diet on the gut microbiome and, in turn, on the development of cardiovascular disease and inflammatory bowel disease. Furthermore, it synthesizes recent research on the effects of high-salt diets on the gut microbiome and the gut immune system. This further elucidates the mechanisms through which a high-salt diet affects the gut microbiome and associated diseases.

**Abstract:**

Salt, or sodium chloride, is an essential component of the human diet. Recent studies have demonstrated that dietary patterns characterized by a high intake of salt can influence the abundance and diversity of the gut microbiota, and may play a pivotal role in the etiology and exacerbation of certain diseases, including inflammatory bowel disease and cardiovascular disease. The objective of this review is to synthesize the effects of elevated salt consumption on the gut microbiota, including its influence on gut microbial metabolites and the gut immune system. Additionally, this review will investigate the potential implications of these effects for the development of cardiovascular disease and inflammatory bowel disease. The findings of this study offer novel insights and avenues for the management of two common conditions with significant clinical implications.

## 1. Introduction

The intestinal microbiota constitute of millions of microorganisms that inhabit the gastrointestinal tract and can interact with the host on a regular basis. These microorganisms include bacteria, fungi, archaea, protozoa, and viruses [1]. The primary capabilities of the intestinal microbiota are the digestion of food, the provision of vital nutrients to the host, defense against different pathogens, and the regulation of the differentiation and growth of intestinal epithelial cells [2]. It is, therefore, crucial to ensure the maintenance of the typical microbiota composition in the intestine to safeguard human digestive and immune health. Recently, several studies have demonstrated relevance between changes in the abundance and diversity of the intestinal microbiome (dysbiosis) and the etiology of a range of diseases, including cardiovascular disease [3], atherosclerosis, hypertension [4], and inflammatory bowel disease [5]. Moreover, it has been demonstrated that additional host factors, such as dietary habits, can influence the composition of the microbiota [1].

Salt, also known as sodium chloride and NaCl, is a vital part of the human diet. The terms “salt” and “sodium” are frequently employed in a manner that suggests an interchangeable meaning [6]. Given the pivotal role played by salt in the human diet, the influence of sodium consumption on human health is being increasingly highlighted, particularly in the context of disease risk. A recent statistical analysis of dietary trends in China revealed that a diet high in salt, and particularly sodium, has become a prevalent feature of the country’s dietary habits [7,8,9]. The World Health Organization (WHO) Healthy Diet Fact Sheet recommends a daily intake of less than 5 g of salt [10]. Nevertheless, despite a reduction in sodium intake in China from 6.3 g/d in 1991 to 4.1 g/d in 2015, the level remains double the tolerated upper limit suggested by the WHO [11]. This high-salt-intake dietary pattern, which is becoming increasingly common, has the potential to negatively impact the health and well-being of the Chinese people [12]. Moreover, over the past few decades, most Americans have consumed sodium more than the recommended levels, which are less than 2.3 g/day according to the American Heart Association’s recommendation [13]. The average American consumes 3.4 g of sodium (or 8.5 g of salt) each day, which is much more than the recommended amounts, according to data from the Food and Drug Administration (FDA) [14]. These results indicate that excessive salt intake is a prevalent phenomenon in numerous countries. Given the intimate connection between diet and our lives, the potential dangers of a high-salt diet are often overlooked. It is, therefore, crucial to conduct research and gain a deeper understanding of the risks associated with a high-salt diet, in order to provide individuals with comprehensive and evidence-based nutritional and dietary guidance. Furthermore, given the established link between diet and gut microbiology, with implications for disease development, several recent studies have explored the impact of a high salt intake on gut microbiology and disease [15], including inflammatory bowel disease [6] and cardiovascular disease [16]. These studies serve to illustrate that a diet high in salt can exert a detrimental effect on the gut microflora, which, in turn, can impair gut health and thereby contribute to the etiology and progression of a number of diseases.

In this review, we place greater emphasis on the effects of a high-salt diet on the gut microbiota. Furthermore, we summarize the disease implications of these microbiological changes, with a particular focus on cardiovascular disease and inflammatory bowel disease. In this context, we discuss the mechanisms by which dietary salt alters the gut flora and the gut immune system, and the impact of this on these two common diseases. This offers insights that can inform the daily dietary management of individuals with the aim of preventing and mitigating the progression of related diseases.

## 2. Dietary Salt’s Effect on the Gut Microbiota

### 2.1. Gut Microbiota and Immune System

The gut microbiota and intestinal immune system interact dynamically, and dietary salt can influence this interaction, meaning it can affect intestine immunity. In the gut microbiota, some beneficial bacteria have been shown to inhibit host immune cells, such as *Lactobacillales*, *Clostridium sensu stricto*, *Prevotella*, and *Alloprevotella* [17]. Additionally, certain harmful bacteria have the capacity to induce the production of inflammatory cytokines in the intestinal tract through interactions with immune cells or the action of their metabolites, thereby contributing to the development of intestinal damage, such as *Escherichia coli* and *Prevotellaceae*, which have been linked to metabolic syndrome and chronic inflammation [18,19,20]. In addition, some conditionally pathogenic microbiota can be considered as symbiotic under normal conditions; however, in instances of inflammation or overgrowth, they may exert a deleterious influence on gut health. For example, *Bacteroides*, *Candida*, *Alistipes*, and *Anaerotruncus* [17]. It is, therefore, essential to maintain a balance between beneficial and harmful bacteria to maintain immune homeostasis and reduce inflammatory responses in the gut.

These intestinal microorganisms exert their effects on the gut and systemic immunity by the means of the metabolites they produce, including short-chain fatty acids (SCFAs), indole, and bile acid (BA) metabolites, thus maintaining the gut and systemic homeostasis. Normal gut microbiota conditions allow for the secretion of these compounds, which inhibit inflammatory cells such as DC, iMac, neurotrophil, Th1, Th2, CD8, and Th17 cells and help to promote the differentiation and function of immunosuppressive cells such as tDC, Treg, MDSC, Breg, tMac, and Tr1 [21]. A reduction in these metabolites may, therefore, have a detrimental effect on the gut’s immune environment. For instance, it is notable that certain pathological conditions, particularly those of an inflammatory disorder, frequently exhibit compromised immune functions, which are often linked to changes in the composition of the gut microbiota and metabolites [22].

In conclusion, microorganisms and their products within the body are important for maintaining intestinal immune homeostasis. Changes in microbial richness and diversity can lead to an imbalance of the gut immune system and be associated with the pathological processes of a number of diseases. This illustrates that certain environmental factors capable of causing microbial alterations may indirectly or directly influence the pathological processes of diseases, thereby giving us potential means for prevention or treatment.

### 2.2. High Dietary Salt and Gut Microbial Composition

The structure and activity of the trillions of microorganisms that reside in the human gut are influenced by both long-term and short-term dietary intake [23]. The direct long-term effects of dietary intake on host physiology can be either beneficial or detrimental, contingent on the specific dietary regimen [24]. High dietary salt intake can lead to a reduction in beneficial intestinal microbiota, which can inhibit the inflammation response. A reduction in the number of bacteria that are beneficial for gut health, such as Lactobacilli [25], has been observed. This reduction has been shown to down-regulate some anti-inflammatory responses and disrupt immune homeostasis in the gut. Furthermore, an increase in certain harmful bacteria has been shown to exacerbate the inflammatory response, thereby promoting intestinal inflammation. This is particularly evident in the case of bacteria that are detrimental to intestinal health, such as some members of Bacteroides [17].

The majority of studies looking at the influence of elevated salt consumption on the gut flora have employed mouse models. A study of microbial community composition indicated that a high salt consumption could enhance the richness of the small intestinal microbiota while diminishing its diversity. Furthermore, a diet with a high salt intake could exacerbate imbalances in the small intestinal microbiota composition, which could ultimately result in an unhealthy state [26]. More particularly, a diet high in salt has been demonstrated to alter the profile of the gut microbiota, resulting in a rapid depletion of *Lactobacillus* spp. in both mice and humans [15]. *Lactobacillus* spp., which belongs to Firmicutes at the phylum level, has been demonstrated to possess the capacity to both prevent and reduce the incidence of inflammatory bowel disease and other inflammatory diseases by modulating the JAK/STAT and NF-κB signaling pathways [25,27]. It can be reasonably inferred that a diet high in salt intake will result in a reduction in the abundance of intestinal *Lactobacilli* spp. Such a reduction will, consequently, give rise to an attenuation in the anti-inflammatory response, which may, in turn, contribute to the development of inflammatory bowel disease. Furthermore, Yan X. et al. demonstrated that a diet with a high salt content led to a notable reduction in beneficial Bacteroidetes such as Bacteroidetes fragilis, which changed the metabolic profiles and eventually led to a high blood pressure [28] (Figure 1).

In contrast, a high salt intake has also been linked to an increased level of certain microbial species. In a more detailed classification, Wang C et al. found that the abundances of *Lachnospiraceae* and *Ruminococcaceae* are greater and that there is a higher ratio of Firmicutes to Bacteroidetes (F/B) in mice fed with a high-salt diet than in those fed with a low-salt diet [29]. Members of *Ruminococcus* have been identified as the primary mucolytic bacterium in patients with Crohn’s disease [30]. Their metabolites have the potential to disrupt the mucus layer, thereby facilitating bacterial translocation and establishing a link with IBD [31]. Furthermore, the F/B ratio is one of the biomarkers used to assess gut health. An alteration in the F/B ratio, whether an increase or a decrease, is indicative of dysbiosis [32].

Furthermore, a pilot study conducted on Wistar rats demonstrated that a diet with a high salt intake resulted in a notable alteration in the composition of the intestinal microbiota [33]. The findings of the pilot study demonstrated that the populations of *Alloprevotella*, *Prevotella 9*, *Allobaculum*, and *Turicibacter* were increased in the high-salt rat cohort. Conversely, the populations of *Prevotella NK3B31* and *Helicobacter* were diminished in the high-salt group. Furthermore, no physiological or pathological alterations were discerned in the high-salt group, suggesting that the alteration in the microbiota composition preceded the emergence of physiological symptoms [33]. In a rat model, Wilck N. et al. found that the relative abundances of *Oscillibacter*, *Pseudoflavonifractor*, *Clostridium XIVa*, *Johnsonella*, and *Rothia* were significantly reduced following eight weeks of a high salt intake, while *Parasutterella* spp. exhibited an increase [15]. The findings of this study indicate that excessive salt consumption results in alterations to the gut microbiome. However, the precise mechanisms through which these microorganisms exert their effects remain to be elucidated.

In conclusion, these findings suggest that an excessive salt intake in the diet induces microbiological changes in the intestinal flora of humans and mice. These changes include a reduction in beneficial bacterial species, which leads to alterations in intestinal immune homeostasis and exacerbates the inflammatory response in the gut. It can be postulated that these changes represent a risk factor for inflammatory or immune diseases of the gut. Furthermore, these microbial alterations offer a novel strategy for the treatment of related diseases.

### 2.3. Interactions between Gut Microbiome and Short-Chain Fatty Acids in High Dietary Salt

The metabolites generated from the microbiota are essential for host–microbe interactions. A high-salt diet can affect the intestinal microbiota and their metabolites. Short-chain fatty acids (SCFAs), such as butyrate, propionate, and acetate, are metabolites produced by the gut microbiota and offer the intestinal epithelium a significant source of energy [34]. SCFAs are generated through the bacterial fermentation of dietary fiber within the gastrointestinal tract [35]. Additionally, SCFAs are vital for maintaining intestinal homeostasis and possess crucial immunomodulatory functions. They work by blocking histone deacetylase (HDAC) and stimulating GPCRs found in intestinal epithelial cells and immune cells to reduce inflammation in the intestinal mucosa [36]. As a conduit between microbes and the host, the intestinal flora can exert influence over host physiology at the intestinal level or via the bloodstream, given the potential for metabolites to cross the barrier between the two [28]. It can be concluded that a high salt intake affects the gut microbes and metabolites, as well as the normal physiological processes that occur in the host.

In a mice model, Hu L et al. found that high dietary salt can directly cause gut dysbiosis, and particularly reduced SCFA production. Their results demonstrated that a diet with a high salt content led to a notable reduction in the proportions of Bacteroidetes (specifically *S24-7* and *Alloprevotella*) and Proteobacteria, while the proportion of Firmicutes (specifically *Lachnospiraceae* and *Ruminococcaceae*) significantly increased. Additionally, the absolute concentrations of acetate, propionate, and butyrate in fecal samples from mice fed with a high-salt diet were found to be reduced in comparison to those of the control group [37]. Firmicutes and Bacteroidetes are the primary butyrate-producing bacteria, while the Bacteroidetes are also responsible for the production of acetate and propionate [36]. Butyrate has been demonstrated to bind to GPR43, thereby activating the production of anti-inflammatory cytokines, including TGFβ and IL-10, and upregulating FOXP3 in Treg cells. Additionally, butyrate has been shown to inhibit histone deacetylase activity, which, in turn, downregulates NF-κβ-mediated inflammatory responses [38]. A reduction in butyrate levels has been demonstrated to result in an increase in inflammatory processes. This indicates that a high salt intake can lead to change in the gut microbiota to decrease the SCFAs that are considered as healthy, which can affect gut immune homeostasis.

In addition, in a multi-omics analysis of the gut microbial community in IBD, Lloyd-Price J. et al. observed a reduction in SCFAs, particularly butyrate, in IBD patients with dysbiosis samples compared to non-dysbiotic samples with IBD [39]. Similarly, another study demonstrated that, in a mouse model, the exacerbation of colitis induced by a high-salt diet in comparison to a normal diet was associated with a reduction in the genera of major SCFAs producers, including *Lachnospiraceae*, *Clostridiales*, and *Oscillospira*. This, in turn, led to a protective effect in the gut, characterized by a reduction in the production of SCFAs, particularly butyrate [40]. It can be postulated that an elevated sodium consumption may exert a diminishing influence on the concentration of SCFAs, particularly butyrate, in mice models. In patients with IBD, a high salt intake may result in a reduction in SCFAs, which could exacerbate the disruption of the intestinal microbiome and, consequently, the severity of the disease.

However, these findings contrast with Bier A. et al. [41], who showed that fecal samples from rat model fed with a high-salt diet and exhibiting hypertension demonstrated elevated levels of SCFAs, particularly acetate, propionate, and isobutyrate, as opposed to butyrate. This suggests the need for greater understanding in different animal models. Furthermore, the study by Bier et al. posited that no particular bacterial strain was responsible for altering the levels of SCFAs [41]. In summary, the evidence for a high-sodium-intake-induced immune system response indicates an imbalance in the intestinal environment or homeostasis.

## 3. The Influence of Dietary Salt on Cardiovascular Disease

### 3.1. Effects of High-Salt Diet on Intestinal Microbiota in Cardiovascular Disease

Cardiovascular diseases (CVDs), including hypertension, atherosclerosis, and heart failure, represent the leading cause of mortality on a global scale [42]. An excessive intake of salt is regarded as a significant risk factor for the development of cardiovascular disease and can affect heart health in various ways. A modeling study conducted by Tan M. et al. suggested that a reduction in sodium intake in China, even at the modest level of 1 g/day, could prevent approximately 9 million CVD events by 2030, of which almost 4 million could be fatal if sustained over the same period [43]. Furthermore, recent studies have demonstrated that an excessive dietary salt intake, particularly the sodium it contains, can elevate the blood pressure by affecting the gut microbiota [15,44] (Figure 2).

The impact of a diet with a high salt content on the development of cardiovascular disease is primarily observed in its effect on blood pressure, which is influenced by changes in the gut microbiota. For example, a systematic review of observational studies revealed that gut microbiota dysbiosis was a common feature in hypertension, including decreased diversity, an altered microbial structure, compositional changes in taxa, and alterations in microbial function [45]. As demonstrated by Wilck N et al. in mouse model systems [15], a high salt intake exerts a detrimental impact on the intestinal microbiome, notably by diminishing the population of *Lactobacillus murinus*. Additionally, it has been observed to enhance the prevalence of Th17 cells and elevate blood pressure. In a similar study, Ferguson JF and colleagues observed that, in both humans and mice, a high-salt diet resulted in a reduction in lactate-producing bacteria, including members of the *Bacilli* class, the *Lactobacillales* order, the *Leuconostocaceae* family, and the *Leuconostoc* genus [19]. These bacteria typically inhibit salt-induced T cell activation and hypertension [15]. Furthermore, the findings indicated that an excessive salt intake was linked to the colonization of bacteria associated with inflammation and metabolic syndrome, including *Lachnospiraceae* and *Prevotellaceae* [15]. Consequently, alterations in these bacterial species result in corresponding modifications in their bacterial-associated functions, which not only facilitate intestinal inflammatory responses, but also impede the inhibitory effects on hypertension.

In addition to directly influencing the composition of the gut microbiota, an elevated salt consumption can also exert an indirect effect on salt absorption pathways, including the sodium-proton exchanger-3(NHE-3). The absence of NHE-3 can result in alterations in the gut microbiological environment and a decrease in blood pressure [46]. Furthermore, a high-salt diet increases the expression of NHE3 at the apical or basolateral membranes in the intestine [47]. In light of these findings, it seems reasonable to posit that a high salt intake may lead to an increase in NHE3, which could, in turn, result in a greater absorption of salt, particularly sodium, as well as alterations in the abundance of gut microbes and an overall rise in blood pressure. Furthermore, a review by Masenga et al. posits that dietary salt may indirectly exacerbate the risk of heart failure by modulating the gut microbiota and contributing to hypertension [48]. In conclusion, a high salt intake can affect the gut microbiology, which, in turn, can lead to hypertension. This alteration in blood pressure can further contribute to heart failure to some extent.

In conclusion, the findings of these studies indicate that an elevated salt consumption influences the gut microbiota, which may, consequently, affect cardiovascular health. It is of the utmost importance to gain a deeper understanding of the relationship between salt intake, the gut microbiota, and cardiovascular disease in order to develop effective strategies to reduce the risk of cardiovascular complications associated with a high salt intake.

### 3.2. Effects of High-Salt Diet on Intestinal Microbiota Metabolites in Cardiovascular Disease

As previously discussed, an elevated salt intake has been shown to influence the composition of various functional gut microorganisms, which, in turn, has been linked to an increased risk of associated cardiovascular disease. Concurrently, a substantial body of research has elucidated the impact of an elevated salt intake on the composition and function of the gut microbiome, and the roles of these alterations in the pathogeneses of associated diseases.

The microbial products most closely associated with the effects of high dietary salt are SCFAs. SCFAs have been demonstrated to serve as a fuel source for the heart in cases of heart failure, thereby enhancing cardiac function. Additionally, they have been shown to exert a protective effect against systemic hypertension and left-sided heart lesions [49,50]. As previously discussed, an elevated salt consumption results in alterations in the composition of the gut microbiota, notably a decline in the prevalence of Lactobacillus, which is known to produce anti-inflammatory mediators, including SCFAs. It can be postulated that a diet high in salt will result in a reduction in the levels of SCFAs, which may, in turn, exacerbate the symptoms experienced by patients with heart failure. In addition, these SCFAs have been demonstrated to increase the production of anti-inflammatory gut hormones in enteroendocrine cells, including GLP-1 and GLP-2 [51], which are both known for beneficial effects against cardiovascular disease [52,53].

In a hypertension model induced by a high-salt diet in Wistar rats, Yan X. et al. observed that the levels of Bacteroides fragilis and its metabolite arachidonic acid in the intestine were reduced by the high-salt diet [28]. This resulted in an increase in intestinal-derived corticosterone production and corticosterone levels in the serum and intestine, which, in turn, promoted an elevation in blood pressure. Additionally, in a rat model, Zheng et al. demonstrated that a high-salt diet can induce the hypertension of liver-Yang hyperactivity syndrome, primarily through its impact on amino acids and their derivative metabolisms [54]. Notably, γ-aminobutyric acid (GABA) was down-regulated in a high-salt diet, which altered the GABA metabolic pathways, promoting hypertension and leading to an increased blood pressure, while glutamate and its derivatives were up-regulated.

In conclusion, studies show that the effects of a high-salt diet on gut microbial metabolites, particularly SCFAs, can have a direct effect on changes in blood pressure and, thus, cardiovascular disease. Nevertheless, studies explicitly identifying the effects of specific SCFA species produced by specific microorganisms on blood pressure remain scarce, underscoring the need for further research in this area. It is also necessary to consider the potential impacts of other metabolites on blood pressure.

## 4. The Influence of Dietary Salt on IBD

### 4.1. Effects of High-Salt Diet on Intestinal Microbiota in IBD

Inflammatory bowel disease (IBD), encompassing Crohn’s disease (CD) and ulcerative colitis (UC), is typified by intestinal inflammation, which is chronic and relapsing. Recently, it has become a significant global health concern with a consistently rising prevalence [11]. Based on projections derived from the 2020 United States Census, it is estimated that approximately 2.39 million individuals in the United States have been diagnosed with IBD [55]. Furthermore, the incidence of IBD in China was 10.04 per 100,000 person-years in 2016 [56].

The etiology of IBD remains incompletely understood [57], but alterations in the gut microbiota have been identified as playing a role in the pathogenesis of IBD [58]. For example, the levels of bacteria such as *Clostridia*, *Ruminococcaceae*, *Lactobacillus*, and *Faecalibacterium prausnitzii* have been found to be lower in patients with IBD, while *Escherichia coli* and *Fusobacterium* species have been found to be higher in the same patients [59]. Furthermore, alterations in the abundance of fungal microorganisms, such as *Candida albicans* strains, which constitute an integral component of the gut microbial ecosystem, have also been observed in the direct effects of a high-salt diet on host immune homeostasis in IBD [60,61]. In conclusion, the altered abundances of these gut microbes represent a significant factor in the pathogenesis of IBD. By comparing the changes in the intestinal microbiota in IBD and high dietary salt, some similarities can be found.

For example, a combined study of sequencing and the proteome conducted by Wang C. et al. revealed that *Ruminococcus* levels rise after a high-sodium diet [29]. Similarly, a multi-omics study conducted by Lloyd-Price J. et al. revealed that the abundance of *Ruminococcus* was elevated in patients diagnosed with UC and CD [39]. Despite the absence of direct evidence indicating that an elevated salt consumption influences the prevalence of *Ruminococcus* and, consequently, impacts IBD, we can hypothesize about the potential involvement of *Ruminococcus* in the association between a high-salt diet and IBD based on the findings of these two studies.

In conclusion, an investigation into the effects of a high-salt diet on the gut microbiota and alterations in the gut microbiota observed in patients with IBD may help to develop a novel therapeutic strategy for the microbiological treatment of IBD and enhance our comprehension of the influence of diet on IBD.

### 4.2. Direct Effects of High-Salt Diet on Host Immune Homeostasis in IBD

The etiology of IBD remains largely unidentified. However, it is understood that a complex interaction between genetic, environmental, and microbial factors, in conjunction with immune responses, is involved. Moreover, a significant body of evidence derived from a multitude of studies substantiates the correlation between an elevated salt consumption and augmented intestinal inflammation [62,63,64,65] (Figure 3).

Kleinewietfeld M. found that an elevated dietary salt intake may potentially serve as an environmental risk factor for the onset of autoimmune disorders by fostering the proliferation of pathogenic Th17 cells [66]. As a CD4 T cell subset, Th17 cells have been demonstrated to play a dual role in the etiology of IBD, primarily exerting a proinflammatory effect. An animal model demonstrated that an elevated intake of salt can prompt the proliferation of Th17 cells’ differentiation within the small intestine, leading to the accumulation of Th17 cells in the intestinal lamina propria (LP) of the distal small intestine and increased IL-17 in both the intestine and plasma IL-17 levels [67]. IL-17, as a Th17-derived cytokine, plays a significant role in the chronic inflammatory processes that can up-regulate the production of pro-inflammatory mediators, which underpin the pathogeneses of autoimmune diseases such as IBD [68]. Furthermore, IL-17 has been evidenced to directly mediate intestinal inflammatory responses, which has been demonstrated to stimulate the release of IL-8 from damaged intestinal epithelial cells, thereby inducing the chemotaxis of neutrophils and Th17 cells to the site of inflammation [69]. Another study revealed that mice fed with a high-salt diet had more IL-17A-producing cells within the LP than mice fed with a standard diet. Additionally, the high-salt diet triggered an intensified intestinal Th17 response while concurrently impeding the functionality of Treg cells [65]. Hernandez et al. discovered that an elevated salt intake impaired the inhibitory function of Tregs in vivo. This is achieved through the dependence on serum glucocorticoid kinase-1 (SGK1), which, in turn, impairs the inhibitory function of Tregs and promotes a Th1-type effector phenotype in Tregs with a high secretion of IFNγ. This can exacerbate autoimmune disease [70]. In conclusion, an excessive salt intake has been shown to result in alterations to the intestinal immune system, primarily through an impact on the equilibrium between Th17 cells and Treg cells. This leads to a reduction in the suppression of inflammation, while simultaneously elevating pro-inflammatory factors. This evidence suggests that a high-salt diet not only exacerbates existing intestinal inflammatory diseases such as IBD, but also increases the susceptibility of the normal gut to intestinal inflammation.

Furthermore, a growing amount of data indicate that a high salt intake also has the ability to activate various cells and pathways to induce Th17 cells, in addition to directly stimulating an increase in these cells. In mouse models, a high salt intake has been demonstrated to exacerbate autoimmune encephalomyelitis and colitis through the p38/MAPK signaling pathway in intestinal lamina propria mononuclear cells (LPMCs). The exposure of intestinal LPMCs to high dietary salt was observed to enhance the expression of IL-17A, RORγt, and TNF-α, which are recognized to exacerbate mucosal inflammation linked to IBD by inducing Th17 cells [71]. Additionally, a high salt concentration has also been observed to promote IL-23 receptor (IL-23R) expression in naïve T cells by a mechanism involving the SGK1, establishing that the IL-23/IL-23R axis plays a pivotal role in the differentiation of pathogenic inflammatory Th17 cells [72]. Furthermore, an increase in dietary salt levels has been demonstrated to induce a pro-inflammatory M1 phenotype in human and murine macrophages [73]. These M1 macrophages are in charge of producing pro-inflammatory cytokines, including IL-12, IL-23, and IL-1β. This leads to the augmentation of Th1- and Th17-cell-mediated immune responses, which, in turn, exacerbates the damage to the intestinal epithelium [74]. In conclusion, it can be stated that a diet high in salt affects the immune homeostasis of the gut. This effect is not a direct cause of IBD, but it is one of the causative factors involved in IBD. This effect may not only be direct, affecting the intestinal cells, but also indirect, through the action of certain cytokines that increase the percentage of Th17 cells and promote inflammation.

In addition to stimulating Th17 and Treg cells and the associated activation of the Th17 cell pathway, a substantial body of evidence from a range of studies indicates that a high-salt diet can also promote inflammation via other immune cells and pro-inflammatory factors. For instance, Qi et al. recently demonstrated that a diet high in salt can also accelerate the progression of IBD by promoting RIPK3-dependent necrotic apoptosis in mouse colonic epithelial cells when compared with those who consuming a normal or low salt diet [75]. Similarly, given the role of IL-17 cytokines in colonic epithelial cells following injury [69], it seems reasonable to hypothesize that an excessive salt intake in the diet not only causes the damage and death of intestinal epithelial cells, but also increases Th17 cells and IL-17 cytokines, thereby exacerbating the inflammatory response at the site of loss and, in turn, contributing to the development of inflammatory diseases of the intestinal tract. Furthermore, an additional study demonstrated that diets high in sodium chloride influence the mucosal immunity of both the colon and, to a lesser degree, the small intestine. This effect is observed through the enhancement of the expression of pro-inflammatory genes, including Rac1, Map2k1, Map2k6, and Atf2. Conversely, the expression of numerous cytokines and chemokines is suppressed, including Ccl3, Ccl4, Cxcl2, Cxcr4, and Ccr7 [40]. Liao Y et al. demonstrated that the transient intake of high levels of salt was observed to elevate histone 3 lysine 4 monomethylation (H3K4me1) at the nuclear factor κB (NF-κB) subunit p65 promoter in rats exhibiting sensitivity to salt. This promoted the transcription of the p65 gene and the subsequent activation of NF-κB, which, in turn, gave rise to a cascade of inflammatory responses in salt-sensitive hypertension [76]. It has been demonstrated that the transcription factor NF-κB plays a crucial role in regulating dysregulated cytokine secretion and signaling mechanisms in intestinal epithelial cells, lymphocytes, and macrophages. Furthermore, NF-κB-p65, which is present in macrophages and epithelial cells isolated from inflamed intestinal samples from IBD patients, has been shown to enhance this process [77]. These findings collectively highlight the complex interplay between dietary salt intake and the intestinal immune system. It seems reasonable to posit that a diet high in salt may exert an influence on the pathogenesis of IBD by affecting this epigenetic change to promote the expression of particular genes.

To conclude, the available evidence suggests that a high intake of salt or sodium can have a direct effect on the immune system balance within the gut, potentially leading to the development of IBD via various routes.

## 5. Treatment

### 5.1. Dietary Therapies for IBD

Given the previously cited data, it is reasonable to assume that dietary practices have a significant impact on the gut microbiome’s makeup and, subsequently, influence the progression of IBD. This finding supports the theory that dietary modification could be a practical means for treating IBD. Furthermore, Hashash JG et al. provided the best dietary advice for people with IBD in a study on nutritional and dietary therapies for patients with IBD that was published in 2024. One important suggestion was that all individuals suffering from the illness should eat a diet with reduced salt [78]. The Exclusive Enteral Nutrition diet, which is used to treat Crohn’s disease, will be discussed in this section as an example.

Exclusive enteral nutrition (EEN) is the most reliable and scientifically supported dietary therapy used to treat IBD [79]. EEN therapy is known as the administration of a comprehensive nutritional formula as the sole source of nutrition over a period of 6–10 weeks [80]. A meta-analysis of pediatric studies compared the efficacy of EEN with alternative treatments, demonstrating that EEN induced remission in up to 80–85% of patients [80]. Subsequently, Kaakoush et al. discovered that EEN results in common and patient-specific alterations in the microbiota of patients with CD, a number of which are correlated with disease activity [81]. These findings indicate that EEN may play a role in the management of IBD. Nevertheless, the precise mechanism exerted on the intestinal microbiota by which EEN induces remission remains a topic of contention. For example, a comparative study conducted in 2014 by Gerasimidis K. et al. revealed a decline in global bacterial diversity and abundance after EEN. This finding is at odds with the pervasive notion that a diverse microbiome is crucial for maintaining optimal health. Specially, the results of this study indicated a notable reduction in *F. prausnitzii* following EEN, and butyrate levels are also significantly reduced on EEN, which challenges the prevailing notion of *F. prausnitzii* and SCFA as protective factors in CD [82].

In a recent publication, Kuang and colleagues discussed the correlation between salt content and EEN. As indicated in their review, the generally low levels of salt present in the majority of EEN regimens are notable. However, the role of salt intake as a mechanistic factor in the remission of CD induced by EEN has yet to be fully elucidated [6]. In conclusion, EEN has been demonstrated to be an effective treatment for IBD, particularly Crohn’s disease. However, further research is required to elucidate the precise mechanism of action.

### 5.2. Dietary Therapies for CVD

The detrimental impact of an excessive salt intake on cardiovascular disease makes it imperative to reduce salt consumption in the daily diet, thereby mitigating the risk of cardiovascular disease by promoting inflammation and blood pressure.

In a study conducted by Ma H and colleagues, a lower frequency of adding salt to foods was found to be associated with a reduced risk of CVD, particularly heart failure and ischemic heart disease [83]. In addition, in a review of the efficacy of salt reduction in the prevention of hypertension and cardiovascular disease, He FJ and colleagues reached the conclusion that a reduction in salt consumption at the population level represents one of the most cost-effective, feasible, and affordable strategies for the prevention of cardiovascular disease [16]. In the United States, a reduction in salt intake to 3 g/day has the potential to prevent approximately 146,000 new cases of CVD and more than 40,000 associated deaths per year [84].

It is, therefore, imperative to identify an efficacious method for the reduction in or replacement of salt. A promising strategy for reducing salt intake is to replace regular salt with salt substitutes. These are made with less sodium and more potassium and have been demonstrated to reduce both blood pressure and the mortality rate from cardiovascular disease [85].

In conclusion, a reduction in the intake of salt on a daily basis has been demonstrated to be an effective method for reducing the risk of developing cardiovascular disease. This method of disease prevention is both cost-effective and efficacious.

## 6. Future Direction

The existing literature has played an important role in elucidating numerous aspects of the relationship between high-salt diets and the gut microbiota. This includes high-salt-induced gut immune responses, microbial ecological dysregulation, and the functional activity of microbial metabolites. Nevertheless, the intricate nature of the human microbiome as a dynamic, interacting system, coupled with the ambiguity surrounding the immediate and long-term impacts of dietary patterns on the gut microbiota, has been made clear. The current state of knowledge permits only a limited range of data to be employed in order to bridge the gap between an excessive salt intake, individual gut microbes, and an altered microbial metabolism and function and the development of cardiovascular disease, as well as inflammatory bowel disease. This indicates the necessity for a more comprehensive methodology to be employed in the study of the microbiome and pathogenic risk factors in the context of a high-salt diet. Furthermore, the majority of researchers continue to utilize mouse models, which underscores the necessity for a more in-depth investigation into the influence of clinical microbiology on humans.

## 7. Conclusions

A substantial body of prior research has demonstrated that diets with elevated salt levels can result in alterations in the composition of the gut microbiota. These changes can result in alterations to a number of normal functions, including the enhancement of the inflammatory response in the gut and an increase in blood pressure. This can, in turn, lead to an exacerbation of CVD and IBD. Furthermore, high-salt diets can also result in a worsening of IBD by directly affecting the immune homeostasis of the gut.

Nevertheless, it is essential to acknowledge the ongoing limitations of current research in these domains. With regard to cardiovascular disease, studies on the associated pathology caused by high-salt diets through effects on gut microbiology have focused on hypertension. However, further research on the direct effects of atherosclerosis and heart failure is still lacking. Furthermore, fewer studies have explicitly investigated the influence of an elevated salt consumption on IBD through its impact on the gut microbiota compared to cardiovascular disease. Nevertheless, the impact of an elevated sodium intake on IBD can be inferred by comparing the shared microbiota under identical conditions.

The study of the effect of a high-salt diet on the gut microbiome and IBD, as well as CVD, is not only of clinical significance, but also provides new guidance for individuals regarding their daily diet.

## Figures and Tables

**Figure 1 biology-13-00674-f001:**
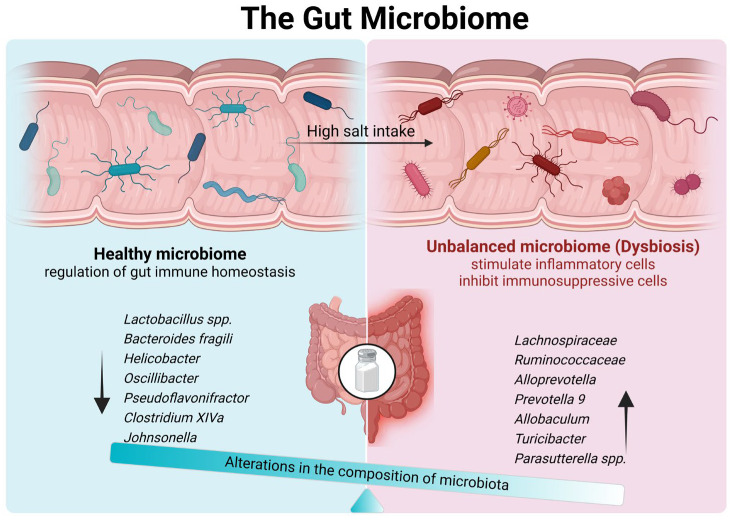
Mechanisms by which high salt intake affects the microbiological composition of the gut. The maintenance of gut microbial homeostasis is integral to the regulation of the immune environment within the gut. An excessive intake of dietary salt can result in alterations to the gut flora, which, in turn, can affect the immune balance within the gut. It is notable that there is a reduction in the prevalence of Lactobacillus and related species, accompanied by an increase in some bacteria. Such disturbances in the flora have been observed to stimulate the activity of inflammatory cells and inhibit the activity of immunosuppressive cells, thereby contributing to an increase in intestinal inflammation.

**Figure 2 biology-13-00674-f002:**
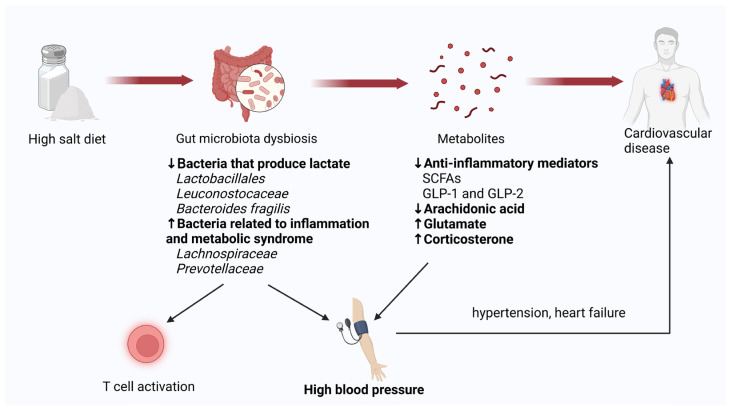
Mechanisms by which high salt intake affects cardiovascular disease by influencing gut microbiota. A diet high in salt can exert an influence on the development of cardiovascular disease by affecting the gut microbiome, which, in turn, affects cardiovascular disease. The effects on microbial composition are primarily a reduction in lactic-acid-producing bacteria and an increase in the abundance of bacteria associated with inflammatory responses. This alteration in bacterial composition subsequently influences the abundance of bacterial-related functions and metabolites, including SCFAs and arachidonic acid. A reduction in these beneficial metabolites activates T cells and elevates blood pressure. Blood pressure is a principal risk factor for cardiovascular disease, leading to hypertension and heart failure.

**Figure 3 biology-13-00674-f003:**
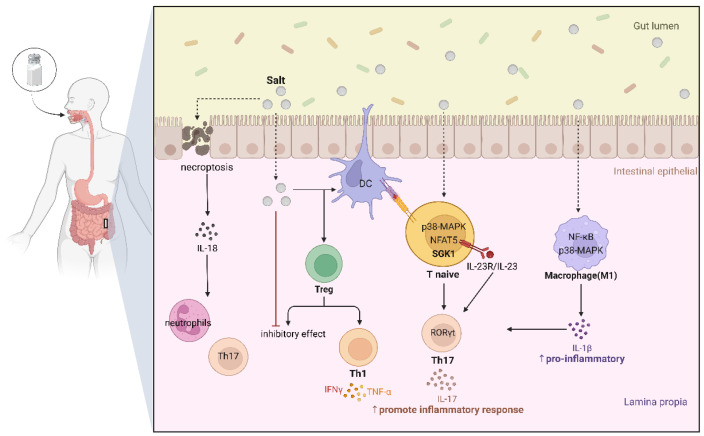
Mechanisms by which high salt intake affects the intestinal immune system. An excessive intake of salt has been demonstrated to exert an adverse effect on the immune system within the gut. Of particular note is its impact on Th17 cells and Treg cells in the LP. A diet high in salt results in an increased differentiation of primitive T-cells to Th17 cells by affecting various pathways, primarily SGK1. This leads to an elevated production of IL17 pro-inflammatory factors and an increased inflammatory response in the gut. In the case of Treg cells, an excess of salt has the effect of suppressing their immunosuppressive function, and converting them into Th1 cells. Furthermore, a diet high in salt can result in the differentiation of macrophages towards the M1 phenotype, which in turn leads to an increase in inflammatory processes. Furthermore, salt can directly induce necrotic apoptosis of epithelial cells, thereby releasing IL18 and further increasing the number of Th17 and neutrophils in the gut.

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
