# Peer review of "High-Salt Diet and Intestinal Microbiota: Influence on Cardiovascular Disease and Inflammatory Bowel Disease"

_biology, 2024, doi:10.3390/biology13090674_

Round 1

Reviewer 1 Report

Comments and Suggestions for Authors

Dear authors a huge subject that must be treated with more rigor. Dealing with the title of your article we expect another structure of the article. We expect a shorter introduction, a wide an complete description of the effects of NaCl on gut microbiota, and maybe with an emphasis on the CVD associated risks that could have impacts on gut health, and finally based on the effects of NaCl on gut microbiota a description of the consequences in relation with IBDs, and a conclusion.

Some remarks:

Salt is not the provider of iodine, it is just a way to complete, see foods are the providers of iodine.

Dealing with the gut microbial composition we were expected more details about the mechanism of action and not only cited references. We would like to understand how it works.

Dealing with the interactions between the gut and SCFA, we would like more details about which bacteria is producing which SCFA. The different SCFA are not equivalent in a physiological way. We would like to have more description between the IBD subjects and the normal ones, like we would like to have more explanations about the NaCl that elevates the SCFA that are considered as healthy.

On line 279, if IBS etiology is not elucidated, please consider that immune homeostasis may participate but not provoke IBS. Or the etiology is elucidated.

It could be also valuable to better discriminate the different type of studies, in vitro, animal, human.

A the end dealing with the treatment part please focus on the NaCl implications, that is the subject of your article.

Reviewer 2 Report

Comments and Suggestions for Authors

Wang et al., have comprehensively reviewed current studies in terms of effects of high salt diet on microbiota and IBD. They also provided insightful ideas and perspective by summarizing main findings in the field and discussed impacts and further directions regarding therapeutic approaches, which is a major research focus currently. Overall, I am happy to recommend this review for the journal consideration.

Meanwhile, I would suggest some minor revisions to make before publication as below:  

1. The scheme of Graphical Abstract shown may be too broad and should be further specified. Suggestions are:

a. define specific topics/ questions in the cartoon, like interactions of high-salt with microbiota (including various species); direct effects on host cells (including epithelia and immune cells)

b. highlight what are unknown questions in each layer of regulations.

2. Line 55: “10.04 per 100,000 years of life in 2016.”  What is the unit of per 100,000 years of life? you mean years of life loss /“YLL”?

3. line59-62: it should also be mentioned that not only bacteria, but also fungi stains like Candida, also have shown IBD association. see PMID: 35859182

4. line88-89: “several studies…” citations are missed here and should have a brief summary sentence of these findings.

5. line100-102: “beneficial” vs “harmful” are subjective terms, need to list example species of each catalog and specifically explain under what conditions? like C.candida normally are commensal but can be toxic if overgrowth/ under inflammation.

6.  line128-130: again, “good”, ”bad” bacteria need to be further defined and specify examples here.

7.  line168-171: Authors mentioned a rat study showing bacteria species changed upon high salt intake, but lack of discussing the biological significance of these changes. What are the consequences? Or it is just from correlation studies with functionality remaining to be characterized?

8. line 206-212: Authors mentioned a conflict of high-salt effects on SCFA from different studies, which is interesting. However, it is important to note that such differences are observed from studies conducted with different contexts such as model organisms (mice vs rat). Indeed, it is common to see the microbiota diversity between species, or even individuals of the same specie. In this case, it is crucial to highlight the study contexts here.

9. line 217: I would suggest the title changed to “Direct effects of high salt on host immune homeostasis in IBD” or similar, based on that the contents discussed in this section are mainly focusing on the direct crosstalk of high salt with host immune cells, unlike previously described high salt- microbiota interactions.

10. line 320-321: lack of citations of individual case study.
